# Numerical Simulation for the Combustion Chamber of a Reference Calorimeter

José Eli Eduardo González-Durán [1],[†] , Marco Antonio Zamora-Antuñano [2],[†] ,
Leonel Lira-Cortés [3],[†] , Juvenal Rodríguez-Reséndiz [4],[*],[†] , Juan Manuel Olivares-Ramírez [5],[†]
and Néstor Efrén Méndez Lozano [2],[†]

[1]   Instituto Tecnológico Superior del Sur de Guanajuato, Guanajuato 38980, Mexico; je.gonzalez@itsur.edu.mx
[2]   Universidad del Valle de México, Querétaro 76230, Mexico; marco.zamora@uvmnet.edu (M.A.Z.-A.);
      nestor.mendez@uvmnet.edu (N.E.M.L.)
[3]   Centro Nacional de Metrología, El Marques 76246, Mexico; llira@cenam.mx
[4]   Facultad de Ingeniería, Universidad Autónoma de Querétaro, Querétaro 76010, Mexico
[5]   Universidad Tecnológica de San Juan del Río, Querétaro 76800, Mexico; jmolivar01@yahoo.com
[*]   Correspondence: juvenal@uaq.edu.mx; Tel.: +52-442-1921200
[†]   Contributed equally to this work.

**Abstract:** This paper focuses on the numerical modeling of the effect of the height of a combustion chamber on the development of a reference calorimeter whose objective is to measure the calorific value of natural gas. The impacts of temperature, velocity, and mass fraction on the exhaust gases were evaluated by varying the height of the combustion chamber. The eddy dissipation concept (EDC) approach was used to model combustion with two different chemical kinetic mechanisms: one with three steps, called the three-step mechanism defined by default in the software used, and second skeletal model, which consists of 41 steps, through the ChemKin-import file with 16 species. The main result of this study is the selection of a combustion chamber height for the reference calorimeter that produces the best performance in the combustion process, which is 70 mm, as well as the main differences in using a three-step mechanism and a skeletal model to simulate an oxy-fuel combustion reaction.

**Keywords:** reference calorimeter; combustion chamber; eddy dissipation concept; oxy-fuel; computational fluid dynamics; thermal modeling

---

## 1. Introduction

Different characterization techniques have allowed us to know or predict the properties of a material and thus assess its usefulness in various applications. Thermal analysis involves a set of technologies that analyze the change in the behavior of a sample when subjected to a scheduled temperature change process in controlled atmospheres: heating, cooling, or isotherms. These analyses establish relationships between the temperature and the physical properties of the material. The result is thermal analysis curves; the characteristics of these curves, such as peaks, discontinuities, or slope changes, are related to thermal events in the sample. Among these different techniques are differential scanning calorimetry (DSC), which can obtain information about material properties and the changes that occur when they are subjected to heat or extraction processes [1,2], the high sensitivity Tian–Calvet heat flow microcalorimeter [3], and adiabatic calorimetry, which is the most accurate method to measure the heat capacity content of the material's enthalpy, which is used to estimate process efficiency. However, in this work we focus on a specific component of isoperibolic calorimetry, which is described in the follow paragraphs.

Currently, natural gas is the third most used fuel worldwide. Measuring the amount of heat released by the complete combustion in air of a specified quantity of gas (on a molar, mass, or volume basis) is the calorific value (CV) [4], which is essential for commercial transactions. Ulbig et al. provided a review of several methods to determine CV [5]. The different techniques available to measure the CV are classified as direct or indirect methods. The direct methods operate by direct combustion, such as the cutlass hammer apparatus. Commercially, some instruments are known as indirect methods, and of which the most used is gas chromatography supported by the ISO 6976 standard [6]. The ISO 15971 standard [4] briefly describes methods to determine CV, such as class 0 mass-basis calorimetry, direct combustion calorimetry, and stoichiometric combustion devices, where the CVs of several pure gases are contained. Methane is the main constituent of natural gas. Measuring the CV is essential because it is used in gas calorimetry as a reference for calibration [7,8]. Today, several institutions around the world [7,9] have developed their own devices operating on the same principle as [8]. Its principal characteristic is high accuracy. The CVs of pure gases that are achieved with this type of apparatus are depicted in [4]. The main components that constitute these devices are:

- The burner provides and mixes the oxidant and the fuel, in addition to generating the flame. The combustion chamber and the heat exchanger are responsible for maximizing the heat transfer of combustion waste gases to its surroundings, usually water.
- The calorimeter vessel contains some fluid, usually water. Its function is to receive and measure the energy generated by the flame and waste gases of combustion, as well as maintain a homogeneous temperature within the contained fluid. The burner, the combustion chamber, and the heat exchanger are immersed in the calorimeter vessel.
- The jacket is another container that includes the calorimeter container and has a uniform, or at least known, temperature.

These calorimeters are known [4] as class 0 mass-basis calorimeters or reference calorimeters. They function under the isoperibolic principle, which consists of observing the temperature increase of a stirred liquid inside the calorimeter vessel while the temperature of the jacket remains constant [8]. Specific measurement results published with reference calorimeters are for pure methane [7,8]. The combustion is carried out to prevent the formation of $NO_x$ and ensure the complete combustion of methane. Unlike using air, oxygen is used as the oxidant. Previous studies [7–9] measured the CV of methane; however, information is lacking about the design and dimensions of the components of the reference calorimeters. In this study, we determined the height of a combustion chamber to optimize combustion performance by analyzing the waste gases exhaust. Almost all engineered and researched products are immersed or make use of some working fluids in their operation or development. This is particularly true of machines for energy generation such as engines, turbines, and renewable energy devices such as wind turbines or wave-energy converters. The ability to model such devices or processes is therefore a key technology, and computational fluid dynamics (CFD) is thus an essential element of digital engineering. Although CFD can loosely be used to denote any computational solution to fluid flow problems, the subject is commonly understood to refer to the resolution of the Euler or the Navier–Stokes equations, or equations derived from these, in two or three spatial dimensions [10]. Other physical effects are often included, either out of interest or necessity. Turbulence is a state of complex, transient, and pseudo-random fluid motion and is almost ubiquitous in energy engineering; it presents several challenges in CFD. Other physical effects are often included, such as chemical reaction and combustion, multiphase flow, free-surface flow, etc. The challenges are both numerical and physical, and several reviews have focused on specific industrial applications or areas of physics [5,7,11–14].

In this study, CFD was used to digitally reproduce the combustion process in the combustion chamber of a reference calorimeter using computational simulation with the aim of determining the optimum height of the combustion chamber of one reference calorimeter that is able to measure the CV of methane. In particular, two chemical kinetic schemes were used and compared: a three-step

mechanism and a skeletal mechanism that considers 16 species and 41 reactions derived from Grimech 3.0. All these analyses were conducted under a combustion approach using the EDC model. The EDC model was developed by [15,16] and describes the chemical process of combustion. The EDC model can incorporate the influence of finite kinetic velocity with a lower computational cost compared to more advanced models, such as the probability density function (PDF) method [17]. Within the EDC model, the influence of turbulent fluctuations on the average chemical reaction rate is considered by referencing the description of turbulence in terms of the turbulent energy cascade. The key variables in the description using the turbulent energy cascade are the speed of energy dissipation and the characteristics of the viscous flow scale (Kolmogorov scale), which only depend on energy dissipation, the kinematic viscosity, and the scale of length, time, and speed of the energy contained in the spectrum range. The results of the EDC model, as reported by [18], accurately follow the experimental data; the EDC approach is the most widely used method for combustion description. Unfortunately, it is also very computationally expensive because the kinetic details are considered [18].

Simulations were performed in Fluent with ANSYS 18.1. The nonlinear governing equations, along with the boundary conditions, were solved with an iterative numerical approximation using the finite volume method [15,17,19,20]. In the solution of the transport equations and the turbulence model, a pressure-implicit with splitting of operators (PISO) algorithm was used for the coupling of pressure and velocity. Figure 1 depicts a schematic diagram of a reference calorimeter [9], focusing on the burner. For the analysis, the burner and combustion chamber were simplified to a 2D axisymmetric model to reduce the computational cost.

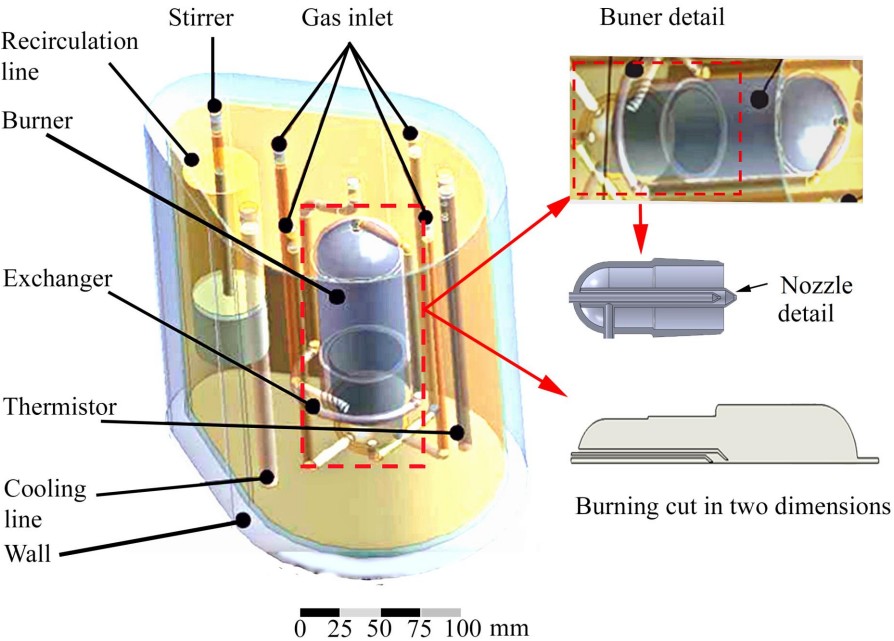

**Figure 1.** Schematic diagram of a reference calorimeter from [9] with details about the nozzle and 2D geometry used in the simulation.

The remainder of this article is structured as follows: Section 2 outlines the related numerical methods and procedures, and covers aspects related to the CFD model established as well as boundary conditions, the kinetics model, turbulence model, and mesh aspects of simulation. Section 3 provides the results obtained from CFD analysis about the model described in Section 2. The information presented demonstrates the differences between the different kinetics models described in this work using graphics of the different burners. Finally, Section 4 provides a discussion and comments on

some relevant aspects of the results, the kinetics model, and the performance of the burner selected in this study.

## 2. Numerical Methods and Procedures

This section describes the model, pressure, and combustion used in all simulations we conducted to find the height of combustion chamber to develop the reference calorimeter. The boundary conditions were established in the simulations. The temperature profile was obtained for two cases that show the differences among the gradients inside the combustion chamber.

The most crucial aspect of modeling kinetics is the number of species and reactions used to simulate the actual combustion phenomenon. However, using many chemical species increases computational time, which implies a better prediction of the real event as opposed to using kinetics with a smaller number of chemical species. The kinetic models were:

- A 3-step mechanism contained by default in the software involving six species.
- The detailed skeletal mechanism with 16-chemical species and 41 reactions, proposed by [18]. Table 1 depicts the oxy-fuel reactions.

**Table 1.** Reactions of the skeletal (SKEL) kinetic mechanism [18].

| Reaction | SKEL | Reaction | SKEL |
|---|---|---|---|
| $H + O_2 \rightarrow OH + O$ | 1 | $CH_2O + OH \rightarrow HCO + H_2O$ | 22 |
| $O + H_2 \rightarrow OH + H$ | 2 | $CH_2O + O_2 \rightarrow HCO + H_2O$ | 23 |
| $OH + H_2 \rightarrow H_2O + H$ | 3 | $CH_2O + M \rightarrow HCO + CH_4$ | 24 |
| $OH + O_2H \rightarrow O + H_2O$ | 4 | $CHO_2 + M \rightarrow HCO + H + M$ | 25 |
| $H + H + M \rightarrow H_2 + M$ | 5 | $CH_3 + O \rightarrow CH_2O + H$ | 26 |
| $H + OH + M \rightarrow H_2O + M$ | 6 | $CH_3 + OH \rightarrow CH_2O + H_2$ | 27 |
| $H + O_2 + M \rightarrow HO_2 + M$ | 7 | $CH_2 + O_2 \rightarrow CH_3O + O$ | 28 |
| $HO_2 + H \rightarrow OH + H$ | 8 | $CH_3 + O_2 \rightarrow CH_2O + OH$ | 29 |
| $HO_2 + H \rightarrow H_2 + H$ | 9 | $CH_3 + H_2O \rightarrow CH_3O + OH$ | 30 |
| $HO_2 + O \rightarrow O_2 + H$ | 10 | $CH_3 + HCO \rightarrow CH_4 + CO$ | 31 |
| $HO_2 + OH \rightarrow H_2O + O_2$ | 11 | $CH_4(+M) + O_2 \rightarrow CH_3 + H(+M)$ | 32 |
| $H_2O_2 + M \rightarrow OH + OH + M$ | 12 | $CH_4 + H \rightarrow CH_3 + H_2$ | 33 |
| $CO + OH \rightarrow CO_2 + H$ | 13 | $CH_4 + O_2 \rightarrow CH_3 + OH$ | 34 |
| $CO + O + M \rightarrow CO_2 + M$ | 14 | $CH_4 + O_2 \rightarrow CH_3 + HO_2$ | 35 |
| $HCO + H \rightarrow H_2 + CO$ | 15 | $CH_4 + OH \rightarrow CH_3 + H_2O$ | 36 |
| $HCO + O \rightarrow OH + CO$ | 16 | $CH_4 + HO_2 \rightarrow CH_3 + H_2O_2$ | 37 |
| $HCO + OH \rightarrow H_2O + CO$ | 17 | $CH_3O + H \rightarrow CH_2O + H_2$ | 38 |
| $HCO + O_2 \rightarrow HO_2 + CO$ | 18 | $CH_3O + OH \rightarrow CH_2O + H_2O$ | 39 |
| $HCO + M \rightarrow H + CO + M$ | 19 | $CH_3O + O_2 \rightarrow CH_2O + HO_2$ | 40 |
| $CH_2O + H \rightarrow HCO + H_2$ | 20 | $CH_3O + M \rightarrow CH_2O + H + M$ | 41 |
| $CH_2O + O \rightarrow HCO + OH$ | 21 | | |

### 2.1. CFD Simulation Set Up

The solver is based on the finite volume method and under the assumptions that the mass, momentum, energy, and species conservation equations are used to calculate pressure ($P$), velocity ($U$), temperature ($T$), and species concentrations ($Y$) [21]. These general conservation equations are as follows:

Continuity:

$$\nabla(\rho U) = 0 \tag{1}$$

Momentum conservation:

$$\nabla(\rho U) = -\nabla P + \mu \nabla^2 U \tag{2}$$

Energy conservation:

$$(\rho C_p)U\nabla T = \nabla(k\nabla T) \tag{3}$$

Species transport:

$$\nabla(\rho U T_i) - \nabla(\rho D_{i,m}\nabla Y_i) = 0 \tag{4}$$

where $\mu$ is viscosity, $C_p$ is heat capacity, $\rho$ is density, $k$ is thermal conductivity, and $D_{i,m}$ is the diffusion coefficient for the $i$th species.

The equations above were described with detail in [22].

The turbulence model chosen for simulation was the $k - \epsilon$ realizable because it is widely used and, according to results of a comparison between other models such as the $k - \epsilon$ standard and the re-normalization of the group (RNG), it performed the best [17,20]. An essential characteristic of combustion models is the ability to efficiently reproduce the physics of the real phenomenon [15,17]. The EDC model was used because its approach is the most commonly used for the description of combustion [17,20]. It produces better results than the flamelet model. The quality of the predictions made using PDF are comparable to that of the EDC predictions [16,17,23,24]. The EDC model, developed by [15,16], incorporates the influence of velocity of finite kinetics with a computational cost that is a bit moderate compared to more advanced models such as the PDF method [15,16,25].

### 2.2. Computational Domain and Mesh

In this work, for the symmetry of the combustion chamber, we used an axisymmetric 2D model to reduce computational time. The model was solved in steady state, which was discretized using ICEM, which is an extension from ANSYS software to generate tetra and hexa meshes. From the independent mesh study, we found that at least three elements were necessary for the minimum part the model, which was 1 mm. This was located just in the flame from the burner, immediately after the exit of the mix of methane and oxygen. The burner consisted of two concentric tubes in which the methane flow circulated in the internal tube and the oxygen flow in the external tube. The geometry of the combustion chamber was a cylinder with hemispherical lid, with a diameter of approximately 45 mm. A simulation was performed for each variation in the height indicated with L in Figure 2 of the chamber: 40, 60, 70, 80, 90, and 100 mm, each kinetic model using the skeletal and the three-step mechanisms. Figure 2 shows a 3D model with a transverse cut plane for three types of combustion chamber evaluated: 40, 60, and 80 mm as an example, where L indicates the height that was changed to analyze its effect on the combustion process in each burner analyzed.

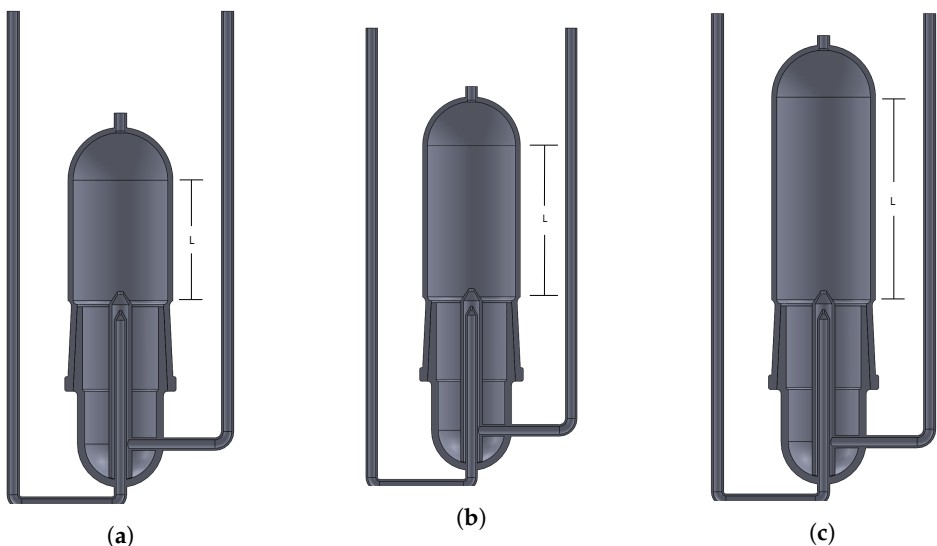

**(a)**　　　　　　　　**(b)**　　　　　　　　**(c)**

**Figure 2.** Burner height of (**a**) L = 60 mm, (**b**) L = 80 mm, and (**c**) L = 100 mm.

### 2.3. Boundary Conditions

The mass flows reported in [9] were used for the boundary conditions, and the mass fractions that were established were 0.96 for methane and 0.9 for oxygen. The inlet temperature for both flows was 25.0 °C. The average values of the results obtained in other simulations were used for the temperature in the different sections of the burner wall, where the interaction between the combustion chamber and a specific amount of surrounding water and the temperature reached by the wall in different sections were established. Table 2 lists the parameters and their values input for the simulations performed in this work; the node value is variable for each burner.

**Table 2.** Parameter and values for the simulations performed.

| Parameter | Values |
|---|---|
| Temperature at border 1 | 571.8 °C |
| Temperature at border 2 | 574.3 °C |
| Temperature at border 3 | 576.3 °C |
| Temperature at border 4 | 587.8 °C |
| Solver | Steady-state, turbulent $k - \epsilon$ realizable model |
| Operating condition | Atmospheric pressure of 10,132 Pa |
| Mass flow rate (inlet-air) | $7.2759^{-6}$ kg/s |
| Mass fraction (inlet-air) | $0.9 \, O_2$ |
| Mass flow rate (inlet-gas) | $8.4^{-7}$ kg/s |
| Mass fraction (inlet-gas) | $0.96 \, CH_4$ |
| Outlet | Gauge pressure of 0 Pa |
| mass fraction (outlet) | $0.9 \, O_2$ |
| Sides | Axisymmetric with $x$-axis |
| Under-relaxation factors | Pressure: 0.3; density: 0.4; body forces: 0.8; momentum: 0.7; species: 0.8; energy: 0.6 |
| Monitor | Mass-weighted average, mass fraction of $CH_4$ |
| Monitor | Mass-weighted average, mass fraction of $O_2$ |
| Residual error | $1 \times 10^{-4}$ for continuity and $1 \times 10^{-5}$ for velocity, $k - \epsilon$, energy, and species |
| Initialization method | Hybrid initialization |
| Iterations | 7500 |
| Nodes | 250,668 nodes for 40 mm burner height; 295,429 nodes for 60 mm burner height; 319,367 for 70 mm burner height; 329,305 for 75 mm burner height; 341,015 for 80 mm burner height; 363,114 for 90 mm burner height; 385,883 for 100 mm burner height. |

Figure 3 depicts the temperature profile for the 40 mm burner height, which was the worst case obtained in this work. The 70 mm burner height in Figure 3b has the best performance among all burners analyzed when the temperature gradients were different for each burner. A flame appeared in the red zone with a temperature of 3436.8 °C.

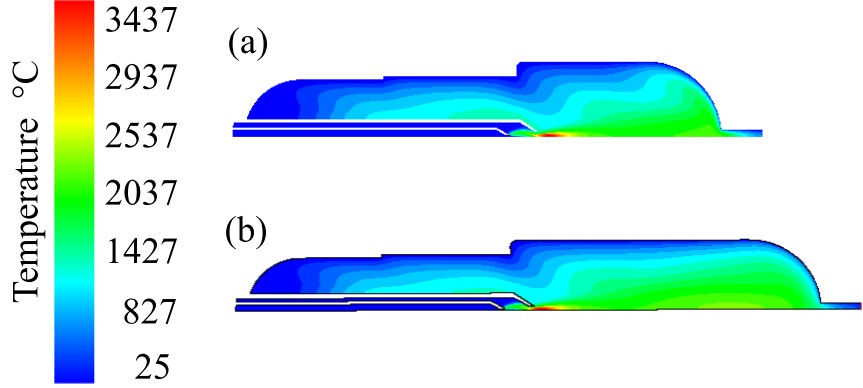

**Figure 3.** Temperature profile for a burner (**a**) 40 and (**b**) 70 mm in height.

The numerical solution is based on the CFD approach in which a finite volume method was implemented to discretize the governing Equations (1)–(4), using commercial software Fluent 18.1.

The steady 2D flow with symmetry at *x*-axis was simulated using the double-precision solver. The pressure–velocity coupling was obtained by the Pressure-Implicit with Splitting of Operators (PISO). The second-order upwind schemes were incorporated for the discretization of the momentum and energy equations. Oxygen and methane concentrations and outlet temperatures were monitored to ensure convergence. With respect to section species, software was used for species transport with volumetric reactions; we selected the ChemKin-CFD chemistry solver with a diffuse energy source. For the mixture properties for the 3-step mechanism, we selected the software default for the methane–air reaction, which includes 6 volumetric species with the EDC for turbulence–chemistry interaction. For all other options, we used the software's own values. For the selection of the mixture properties for the skeletal mechanism, the software has an option to import the ChemKin mechanism. Here, we sectioned two files: one for kinetics input and the other for the thermodynamic database for transport properties. Both files were in format .txt. Using this process, all information required to describe the reaction process was loaded into the simulation.

The domain was meshed with tetra and hexa elements using ICEM software; a fine grid was used near the reaction zone where high gradients were expected. The grid independence was studied using variations of 50,000 nodes in each burner for meshing until finding insignificant variations observed in the monitors used. Therefore, to reduce computational efforts, the nodes that were used for the meshing of the geometry of each burner are listed in Table 2. Convergence was reached because, in our monitors, the calculated values presented a variation of less than 3% in addition to a residual error, as shown in Table 2. Figure 4a depicts the 2D axisymmetric model meshed and in part Figure 4b provides a close-up of the most critical part: the zone where the combustion process occurs.

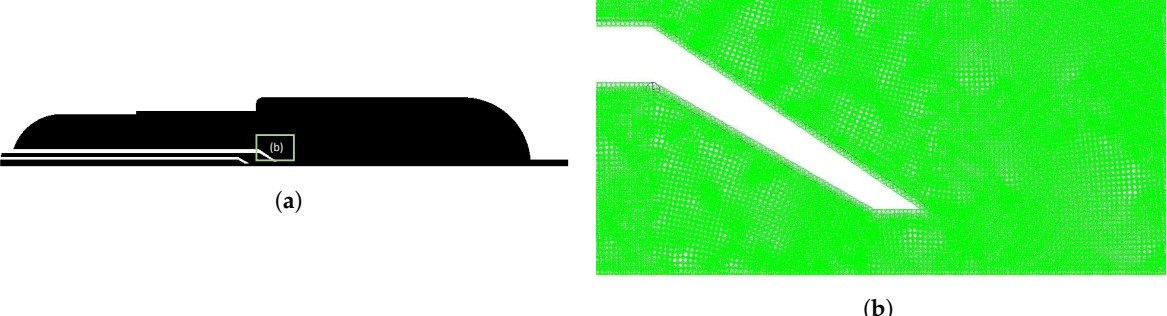

(a)

(b)

**Figure 4.** Mesh (**a**) from entire model and (**b**) a close-up of the critical zone.

## 3. Results

This section presents a comparison of the results obtained in the axial direction when using each kinetic model where "3 STEP" indicates for the three-step mechanism and "SKEL" indicates the detailed skeletal mechanism.

The mass-weighted average values at the output of each burner are shown in Figure 5a for temperature; Figure 5b for maximum temperature, including mean adiabatic flame temperature; Figure 5c for velocity; Figure 5d for the mass fraction of methane; and Figure 5e, Figure 5f, Figure 5g, and Figure 5h for oxygen, carbon monoxide, carbon dioxide, and water vapor, respectively.

The end of this section outlines the elements that were considered for the selection of the optimum height of the combustion chamber that is optimal for the development of the reference calorimeter.

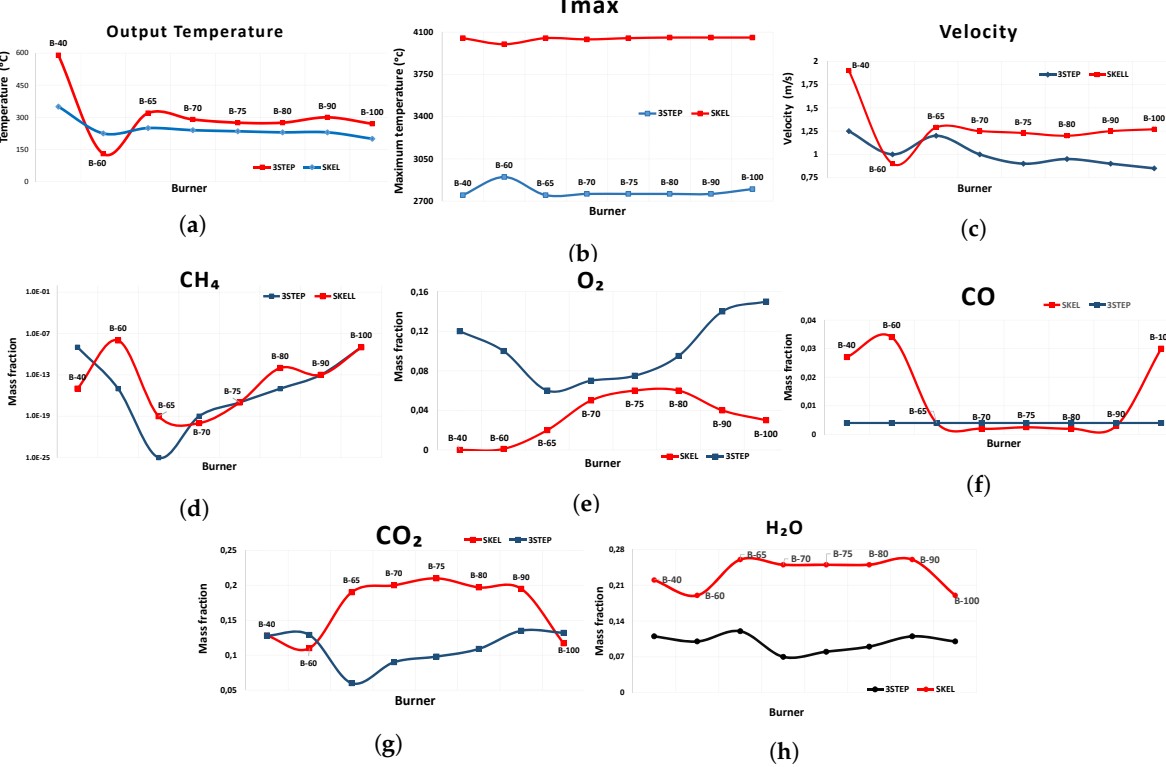

**Figure 5.** Comparison of skeletal and three-step mechanisms: (**a**) Output temperature, (**b**) maximum temperature, (**c**) velocity, (**d**) mass fraction of methane, (**e**) mass fraction oxygen, (**f**) mass fraction of carbon monoxide, (**g**) mass fraction of carbon dioxide, and (**h**) mass fraction of water vapor for the burners evaluated.

*3.1. Comparison of Values at the Burner Exit*

The mass-weighted average values that were obtained at the output of each combustion chamber and for each reaction mechanism used are shown for the three-step and skeletal mechanisms. Its identification is B-40, where *B* refers to the burner, which is related to the assembly of the burner and combustion chamber, and the number represents the height in mm of the combustion chamber. In this section, the burner evaluation results for 65 and 75 mm were added because the 70 mm height produced the lowest amount of methane at the output compared with the other heights. The values for the output temperature, as seen in Figure 5a, were similar except for B-40 because this combustion chamber is shorter, and the heat exchange area is too small. Hence, the temperature at the outlet was higher than for the other burners. B-70 had the smallest deviation in the value between both kinetic models compared to the other burners.

We found a large difference in Figure 5b for the maximum temperature values, which is corroborated in others works because the three-step mechanism produces unrealistic temperatures due to its capacity to control the chemical kinetics. The smallest discrepancy between the two kinetic models was again found for the B-70 burner; the maximum difference was observed for B-80. In output temperature, the B-70 values were closer and the B-40 values were the farthest apart. In this case, the tendencies in both kinetics models agreed.

Figure 5c demonstrates the velocity at the outlet of each burner for the three-step and the skeletal mechanisms. Similar to the decreasing trend in temperature, B-40 had the most and B-70 had the least discrepancy between the two chemical kinetic models.

Again, B-70 had the smallest difference in the mass fraction value of methane from the three-step mechanism. The biggest discrepancies were noted for B-40 and B-60, as well as B-65 between the values produced by the two kinetic models, as shown in Figure 5d.

The results for B-70, B-75 and B-80 for both kinetic models for oxygen were very similar. The values with higher discrepancy were found for B-40, B-60, B-90, and B-100, as shown in Figure 5e.

Figure 5e depicts a higher oxygen mass fraction at the outlet of each burner for the three-step mechanism. This indicated that the three-step mechanism over-predicts oxygen mass fraction values.

For CO, B-70, B-75, B-80, and B-90 had the same results; the most significant differences were identified for B-100, B-60, and B-40, as shown in Figure 5f.

Figure 5g shows the mass fraction of carbon dioxide at the outlet of each burner for the three-step and the skeletal mechanisms. The averaged CO values were the opposite of $CO_2$ as shown in Figure 5g. This occurred because the $CO_2$ molecules that were dissociated formed CO and O in the 40, 60, and 100 mm burners where the CO concentrations were higher than in others.

The differences were noticeable for all burners with respect to the $H_2O$, among the three-step and the skeletal mechanisms, as seen in Figure 5h, indicating that the three-step mechanism under-predicts the mass fraction of $H_2O$.

### 3.2. Selecting the Burner for the Reference Calorimeter

We used the results calculated by the skeletal mechanism to choose the burner with the best combustion performance, because the literature showed that the reduced mechanisms have limitations for handling non-real temperatures [17,20,26] and because simplified mechanisms do not always consider dissociation reactions [17,20,27,28]. The numerical EDC results agreed well with the experimental data according to [18]. In this work was initially established only to consider the least amount of unburned fuel at the exit as a criterion for burner selection. However, the relationship between CO and $O_2$ must be analyzed because CO reflects combustion efficiency [13,26,29].

Figure 6 shows that the least amount of methane at the output was produced by B-70. This could be the criterion used to select the burner with the best combustion process performance. Instead, the additional parameters to be taken into account are described below.

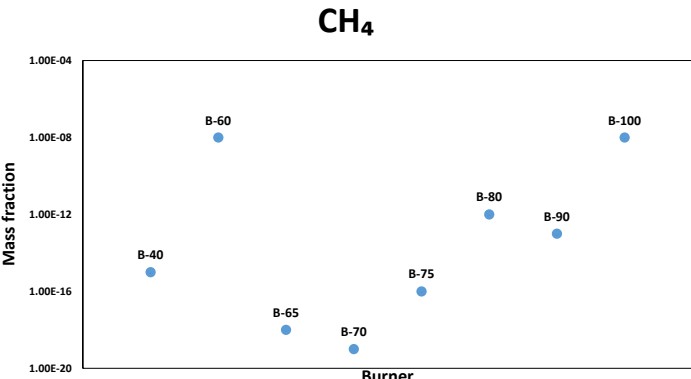

**Figure 6.** Methane mass fraction at the outlet of each burner calculated by the skeletal reaction mechanism.

However, Figure 7a–e depict the species mass fraction of $O_2$, $H_2O$, $CO_2$, and CO for B-40, B-65, B-70, B-75, and B-80 with the aim of considering other parameters to guide the choice of the best burner to develop the reference calorimeter. In the graphs, the *x*-axis starts from inlet of oxygen and methane with 0. At 70 mm, an abrupt change can be observed in the behavior of all the species because this is where the flame was present. The rest of values on the *x*-axis show the behavior with increasing height in each burner evaluated computationally. Until the end, we were able to observe other changes in the curves of species due to the hemispherical geometry at the end of the burner.

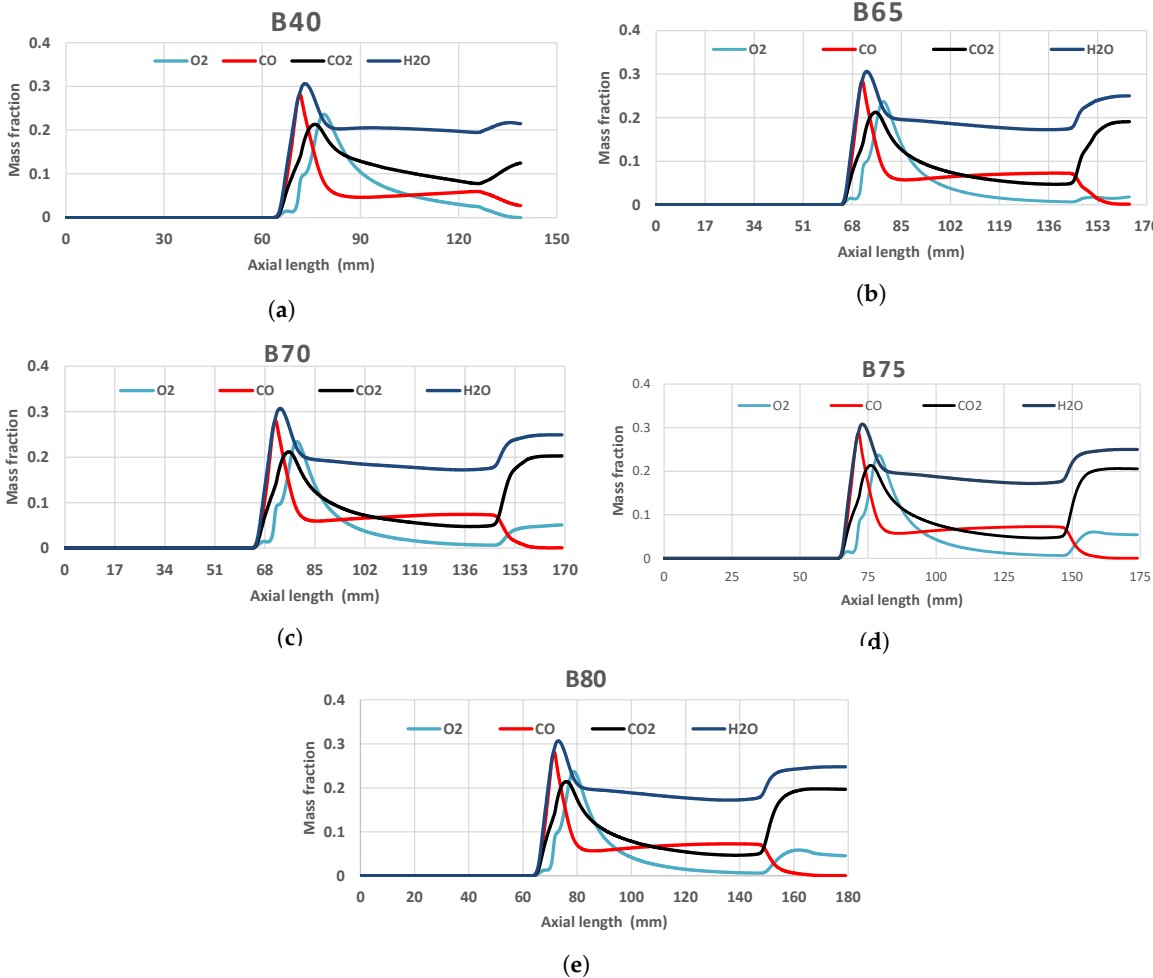

**Figure 7.** Variation in species mass fraction profiles with the skeletal mechanism along the *x* direction; the symmetry line for the evaluated (**a**) 40, (**b**) 65, (**c**) 70, (**d**) 75, and (**e**) 80 mm burners.

Figure 7a shows the mass CO and $O_2$ fraction at the outlet of the burner with a height of 40 mm for the skeletal reaction mechanism. A drop was observed for CO and $CO_2$ at the output of B-40 between 120 and 130 mm, which represents inefficient combustion. Figure 7b shows the CO, $CO_2$, $H_2O$, and $O_2$ mass fractions at the outlet of the burner with a height of 65 mm for the skeletal reaction mechanism. Figure 7b shows that oxygen barely separated from CO at the end of the combustion chamber around 150 to 160 mm for B-65, which resulted in inefficient combustion.

Figure 7c and Figure 7d indicate the CO and $O_2$ mass fractions at the outlet of the 70 mm and 75 mm burner heights, respectively, for the skeletal reaction mechanism. Figure 7d shows that the $O_2$ at the output remained constant, and the CO at the output was lower for B-70 than for B-75. As such, the best option was B-70, as seen in Figure 7c, because the CO is practically 0 and the oxygen value has a positive slope.

Figure 7e describes the CO, $CO_2$, $H_2O$, and $O_2$ mass fractions at the outlet of the 80 mm burner height for the skeletal reaction mechanism. B-80 exhibited the same behavior as B-70. However, the CO value was much lower than the output in this combustion chamber. The $O_2$ decreased, as shown in Figure 7e.

## 4. Discussion

The changes in the geometry of the burner, which acts as a reactor, significantly affect the reaction process. The last changes in the graphs of mass fraction occur the hemispheric section, which begins

in the last 20 or 30 mm in Figure 7a–e. Here, another reaction process between the products begins that is independent of temperature because, at the exit of the chamber combustion, practically all the burners except B-40 reach approximately 500 °C. However, the curves of CO, $CO_2$, and $O_2$ are different. Therefore, the length between the combustion chamber and its geometry significantly affects the performance of the combustion chamber.

Maximum temperatures, as established by [30], are unreal for global mechanisms such as the three-step mechanism considered here. However, the temperatures at the exit were under-predicted by the three-step mechanism with a maximum difference of 40.19% for the 40 mm height burner and a minimum of 20.88% for the 70 mm chamber compared with the skeletal method, although both mechanisms described the same decreasing trend from B-40 to B-100. The velocity retained the same decreasing trend as did the temperature, which was under-predicted by the global mechanism with a maximum difference of 35.62% for the 40 mm chamber and a minimum of 16.08% for the 70 mm chamber [26].

The mass fraction of $O_2$ was over-predicted in all cases by the three-step mechanism, and the maximum deviation at the exit was 99.99% for the 100 mm height chamber and a minimum of 14.15% for the 70 mm chamber height.

The three-step mechanism under-predicted the mass fraction and showed constant behavior throughout the combustion chamber for CO, unlike the skeletal mechanism, which demonstrated an increase in the fraction of CO in the section where the hemisphere begins. The values showed a maximum deviation of 91.18% at the exit of the 100 mm chamber and a minimum of 73.64% for the 70 mm chamber.

We observed minimal variation in the mass fraction for the 40 mm chamber at the output with a value of 4.83% for $CO_2$. However, the opposite was found for the 65 mm chamber, with an amount of 69.92% at the output. In general, the value in the initial section of the burner was over-predicted by the three-step mechanism at the maximum value, as displayed in Figure 5b,e. However, the three-step method under-predicted at the exit of each combustion chamber. For $H_2O$, the 70 mm chamber height presented the most substantial deviation at 72.07%, and the smallest difference in the mass fraction at the exit was produced by the 100 mm height combustion chamber with a value of 50.07%.

The CO, through the three-step mechanism, showed consistent behavior despite the change in height for each burner. However, for the skeletal device, we observed positive slopes in all cases and very drastic variations tending to zero at the beginning of the semi-hemispheric part, as shown in Figure 5f.

If a combustion chamber had to be selected using the three-step reaction mechanism, the decision would be between the 70 and 80 mm heights due to their low methane content and small CO, which implies efficient combustion.

A similar result is provided when using the skeletal mechanism; the decision is between the 70 and 80 mm chambers. According to the previous analysis, although the three-step kinetic model is considered less accurate than the skeletal one, the same result can be obtained qualitatively from both models. From the graphs of CO and $O_2$ for B-70, B-75, and B-80, the combustion chamber with a height of 70 mm was selected as the best option for the development of the calorimeter because the mass fraction of methane at the exit was the lowest which means the least amount of unburned fuel of all the heights evaluated. B-70 produced an increase in $O_2$ concentration and a decrease in CO concentration, which coincide with the optimal combustion conditions.

The literature established that combustion is more efficient when $CO_2$ is maximized at the outlet. Therefore, when evaluating these values in the burners, according to the skeletal combustion, we found that the 75 mm chamber height produces the highest $CO_2$ values. However, the variation was insignificant for the 70 mm chamber, being only 1.4% of the value obtained through the skeletal mechanism. In summary, we evaluated the five most essential parameters at the exit of each chamber: combustion, lower fuel, CO, higher oxygen, and $CO_2$, as shown in Figure 8.

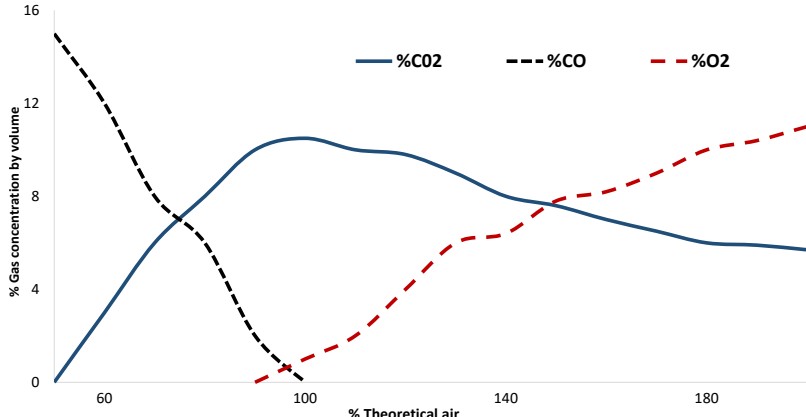

**Figure 8.** Combustion gas concentrations as a percent of the theoretical combustion of air.

Therefore, all the essential parameter averages are shown in Table 3. The distance optimum in the combustion chamber proposed by this work with base in previous analysis is the height of 70 mm.

**Table 3.** Comparison of the most critical parameters for the selection of the optimum combustion chamber.

| Burner | $CH_4$ | CO | $O_2$ | $CO_2$ | $T_{out}$ |
|---|---|---|---|---|---|
| B-70 | $1.81^{-20}$ | 0.00073 | 0.05152 | 0.2027 | 285.95 |
| B-75 | $3.96^{-17}$ | 0.00041 | 0.05388 | 0.2054 | 267.48 |
| B-80 | $3.32^{-13}$ | 0.00036 | 0.04496 | 0.1963 | 263.3 |
| $V_{min}$ | B-70 | B-80 | B-80 | B-80 | B-80 |
| $V_{max}$ | B-80 | B-70 | B-75 | B-75 | B-70 |

## 5. Conclusions

In this work, we studied the combustion performance in a burner of a reference calorimeter using two models of the reaction mechanism using commercial software. We identified that the optimum combustion in the specified burner is a height of 70 mm.

The three-step mechanism produced close results for the $CH_4$ mass fraction, as shown in Figure 7c, and over-predicts the oxygen mass fraction in all the burners evaluated, as shown in Figure 7d. The three-step mechanism does not provide any reliable results for CO, and under-predicts the $CO_2$ mass fraction except for the 60 and 100 mm burners. This situation was also found for the $H_2O$ mass fraction with all values obtained from simulation for the burners evaluated. The three-step mechanism was also found to calculate unrealistic adiabatic flame temperatures.

According to the study, we observed two reaction zones from mass fraction curves in the burners: the first at the location of the flame and the second in the hemispherical section. It occurred because the curves presented changes in their behavior. Therefore, the combustion reaction is affected not only by temperature and pressure but also by the geometry of the combustion chamber.

CFD was used as a tool to design and optimize the combustion chamber for the reference calorimeter under development. For this work, the oxy-fuel combustion was modeled by a skeletal mechanism with 16 chemical species and 41 reactions, since the experimental activities are expensive and risky.

The turbulence–chemistry interaction of the simulated combustion reactions was solved by the EDC model. The EDC model can incorporate detailed chemistry in turbulent combustion, which makes it attractive for simulating a wide range of combustion systems. The analysis performed in this study demonstrated that the EDC model can provide more accurate predictions of the temperature and gas emissions of gas-phase combustion, especially slow-formation pollutant emissions, such as CO, and predictions of species and temperature fields under conditions of weak and high-turbulent flow if a detailed kinetic scheme is used. If this is not the case, the simplified kinetic scheme will need

to be improved. This suggests that it is possible to apply EDC principles to model the flow field in other areas such as industrial biomass furnaces, which cover a variety of flow regimes [1,28,30].

Therefore, in future work, we plan to complement this work using experiment to compare the results and find possible errors.

**Author Contributions:** Conceptualization, J.E.E.G.-D. and L.L.-C.; methodology, M.A.Z.-A.; software, J.M.O.-R.; validation, J.R.-R., N.E.M.L., and J.E.E.G.-D.; formal analysis, J.R.-R.; investigation, M.A.Z.-A.; resources, N.E.M.L.; data curation, J.R.-R.; writing—original draft preparation, J.E.E.G.D.; writing—review and editing, J.R.-R.; visualization, J.M.O.-R.; supervision, M.A.Z.-A.; project administration, J.R.-R. All authors have read and agreed to the published version of the manuscript.

**Funding:** This research was funded by the Consejo Nacional de Ciencia y Tecnología (CONACYT), PRODEP and UVM.

**Conflicts of Interest:** The authors declare no conflict of interest.

## Abbreviations

| | |
|---|---|
| $2D$ | Two dimension |
| B-40 | Burner of 40 millimeter height |
| B-60 | Burner of 60 millimeter height |
| B-65 | Burner of 65 millimeter height |
| B-70 | Burner of 70 millimeter height |
| B-75 | Burner of 75 millimeter height |
| B-80 | Burner of 80 millimeter height |
| B-90 | Burner of 90 millimeter height |
| B-100 | Burner of 100 millimeter height |
| H | Monoatomic hydrogen |
| °C | Celsius degrees |
| CENAM | Centro Nacional de Metrologia |
| $D$ | Diffusion coefficient |
| $c$ | Concentration of chemical species |
| O | Monoatomic oxygen |
| $O_2$ | Diatomic oxygen |
| OH | Phenol |
| $H_2O$ | Water |
| $HO_2$ | Hydrogen dioxide |
| CO | Carbon monoxide |
| $CO_2$ | Carbon dioxide |
| HCO | Aldehyde |
| $CH_2O$ | Formaldehyde |
| $CH_3$ | Radical free methyl |
| $CH_3O$ | Methoxide |
| $CH_4$ | Methane |
| $S$ | Source term |
| s | second |
| kg | kilogram |
| RNG | Re-normalization of the Group |
| EDC | Eddy Dissipation Concept |
| CFD | Computational Fluid Dynamics |
| ANSYS ICEM | extension of the meshing capabilities in ANSYS Meshing |
| DSC | Differential scanning calorimetry |
| m | Meter |
| m/s | meter per second |
| mm | millimeter |
| K | Kelvin degrees |
| NOx | Nitrous oxides |

| CV | Calorific value |
| P | Pressure |
| PDF | Probability Density Function |
| PISO | Pressure-implicit with splitting of operators |
| SKEL | Skeletal |
| T | Temperature |
| Pa | Pascal |
| $\rho$ | Density |
| U | Velocity |
| u | Internal energy |
| $x_i$ | Chemical specie $i$ of reaction |
| Y | Species concentrations |
| $\mu$ | Viscosity |
| $C_p$ | Heat capacity |
| $V_{min}$ | Minimum velocity |
| $V_{max}$ | Maximum velocity |
| $T_{out}$ | Outlet temperature |
| k | Thermal conductivity |
| $D_{im}$ | Diffusion coefficient for the $i$th species |

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
