# Peer review of "Numerical Simulation for the Combustion Chamber of a Reference Calorimeter"

_processes, doi:10.3390/pr8050575_

Round 1
Reviewer 1 Report
The paper describes and reports on a numerical simulation of a gaseous combustion calorimeter related to DSC measurements. While the numerical investigation is reported, its results is not reported in terms of the observed parameters and how they relate back to calorimetric observations of thermodynamic properties of the reaction. The presentation of the analysis and the results need to be improved. For example, Figure 8 indicates an axial length, but that length is not identified on the schematic sketches of the device, Figures 2-4. I found it confusing to have burner dimensions in mm and the axial dimension is tenths of a meter. The authors should insure a consistent set of axial dimensions, identify their definitions and include them on all graphs. For example, figures 6 and 7 seem to plots of different burner lengths, but they are connected by continuous lines. It would be better to report the individual data as points rather than connecting them with a curve. All the figures need to be larger to be more readable.
The simulation is based on two reaction models for this combustion process. How these two models are integrated with Fluent is not discussed or outlined. What parameters are used in the simulation package. This is not a trivial procedure and is an important feature of their analysis. It should be included in the paper.
The summary of the model used, Eqs. 1-5 should be checked by the authors. For example, I would expect that an energy released term or enthalpy of formation term would be included in eq. 3. The results of the analysis suggest that it was included, but Eq. 3 is not consistent with the results. Eq. 4 and 5 seem to be dimensionally incorrect. The authors should check these equation and the definition of the terms.
The english in several parts of the paper is awkwardly worded, lines 60-74, 143-148 are some examples. There are many missing words and missing information under funding.
The author should state earlier and more clearly that they are using a 2-D axisymmetric model rather than referring to it as a 2D model in many places.
A discussion of the difference in the number of node volumes used as reported in Table 2 and that used in the Fluent simulation should included. Also, was convergence for the number nodes used investigated?
In lines 303-308 the authors refer to a sensitivity analysis. I did not find the details of this analysis in the paper. This is important information in evaluating the accuracy of their simulation and should be included in detail.
Lines 326-332 in the conclusion refer to DSC, but the paper is more of a CFD simulation and does not address DSC factors. These lines should be omitted or the authors should relate their results to the DSC measurements. I would have liked to have seen a discussion of the different product compositions as illustrated in Figure 8 and its effect on the measurement of the DSC parameters, ie the heat of combustion, the changes in specific heat as mentioned in the introduction. This connection of the numerical results to the DSC measured parameters is missing from the paper.
The presentation in many places needs to be improved in order for the accuracy and usefulness of the reported simulation to be fully realized. The reported simulation is not easy and the inclusion of different reactions is potentially an important contribution to the design of calorimeters.
Author Response
Reviewer 1
The paper describes and reports on a numerical simulation of a gaseous combustion calorimeter related to DSC measurements. While the numerical investigation is reported, its results is not reported in terms of the observed parameters and how they relate back to calorimetric observations of thermodynamic properties of the reaction. The presentation of the analysis and the results need to be improved.
Thank you for your kindly observation. The entire paper was deeply reviewed and several points were modified along all sections.
For example, Figure 8 indicates an axial length, but that length is not identified on the schematic sketches of the device, Figures 2-4.
Thank you for your comment. We add the variable “L” to identify the value which represents the height, changed in each simulation., 40, 60, 80, etc.
I found it confusing to have burner dimensions in mm and the axial dimension is tenths of a meter. The authors should insure a consistent set of axial dimensions, identify their definitions and include them on all graphs.
Thank you to your kindly observation. It was a mistake, to correct this was changed all the units in (m) to (mm), including graphs
For example, figures 6 and 7 seem to plots of different burner lengths, but they are connected by continuous lines. It would be better to report the individual data as points rather than connecting them with a curve. All the figures need to be larger to be more readable.
Thank you for your comment. In Figures 5 and 6 we have to connect points to analyze the trend in the behavior of varying the height of the burner.
In Figure 7 we are showing the behavior of calculated mass fraction of Oxygen, CO2, CO, and H2O in the line of symmetry and how these values of species are changed along with the height of each burner. Unlike Figures 5 and 6 where is the calculated value at the burner output.
The simulation is based on two reaction models for this combustion process. How these two models are integrated with Fluent is not discussed or outlined. What parameters are used in the simulation package. This is not a trivial procedure and is an important feature of their analysis. It should be included in the paper.
The parameters used in the simulation are shown in Table 2 and we added the follow information:
(line 171 to 179) With respect to section “species” in software was used species transport with volumetric reactions And selected de chemkin-CFD solver to chemistry solver with a diffusion energy source. In mixture properties for the 3 step mechanism was selected the option by default in software to reaction methane-air which includes 6 volumetric species with the EDC for turbulence-chemistry interaction, all other options were taken from the software's own values. In the case to the skeletal mechanism for the mixture properties section in software has an option to import chemkin mechanism import, in this section were sectioned two files, one to kinetics input and another one the thermodynamic database for transport properties, these both files are in format .txt. From this way all information required to describe the reaction process is loaded in the simulation.
The summary of the model used, Eqs. 1-5 should be checked by the authors. For example, I would expect that an energy released term or enthalpy of formation term would be included in eq. 3. The results of the analysis suggest that it was included, but Eq. 3 is not consistent with the results. Eq. 4 and 5 seem to be dimensionally incorrect. The authors should check these equation and the definition of the terms.
Thank you for your kindly observation. Equation 5 was deleted because is the same that equation 4. We add the source where shows a better description of the equations. Eq. 1 to Eq 3 was checked and compared against reference Tahir, F.; Ali, H.; Baloch, A.A.; Jamil, Y. Performance Analysis of Air and Oxy-Fuel Laminar Combustion in a Porous Plate Reactor. Energies 2019, 12. doi:10.3390/en12091706.
The english in several parts of the paper is awkwardly worded, lines 60-74, 143-148 are some examples. There are many missing words and missing information under funding.
Thank you for your appreciable recommendation. We used in the latest version of the manuscript the English service of MDPI.
The author should state earlier and more clearly that they are using a 2-D axisymmetric model rather than referring to it as a 2D model in many places.
Thank you for your observation, taking into account your comment, it was mentioned since the introduction section, (line 99 )
A discussion of the difference in the number of node volumes used as reported in Table 2 and that used in the Fluent simulation should included. Also, was convergence for the number nodes used investigated?
Thank you for your comment. This information was added in boundary conditions section in lines, (184 to 188)
In lines 303-308 the authors refer to a sensitivity analysis. I did not find the details of this analysis in the paper. This is important information in evaluating the accuracy of their simulation and should be included in detail.
Thank you for your observation. Description of the analysis for mesh and results was added in boundary conditions section in lines, (180 to 188)
Lines 326-332 in the conclusion refer to DSC, but the paper is more of a CFD simulation and does not address DSC factors. These lines should be omitted or the authors should relate their results to the DSC measurements. I would have liked to have seen a discussion of the different product compositions as illustrated in Figure 8 and its effect on the measurement of the DSC parameters, ie the heat of combustion, the changes in specific heat as mentioned in the introduction. This connection of the numerical results to the DSC measured parameters is missing from the paper.
Thank you for your appreciable comment. We include somewhat of information on DSC to show an example of other types of calorimeters but our work is focused on the burner for developing the reference calorimeter. To avoid confusion with the mention of DSC calorimeter we decide to delete all paragraphs relational in the conclusion section and we modified in the introduction section the information about DSC.
The presentation in many places needs to be improved in order for the accuracy and usefulness of the reported simulation to be fully realized. The reported simulation is not easy and the inclusion of different reactions is potentially an important contribution to the design of calorimeters.
Reviewer 2 Report
Overall the presentation is too wordy, not to the point. The right scientific wordings have not been used to explain the technical aspects. The manuscript requires significant technical reformatting, systematic presentation approach. The English language, readability needs significant improvement.
Discussion and conclusion is a bit haphazard and jumbled. This needs a more systematic, organized scientific approach.
The reader is lost about what exactly the authors are trying to discover or invent.
Line 1: “effect of height” of what?
L 7: “woks” ?
Lines 63-70, too much text is spent on explaining the advantages of CFD.
Figure 2 caption, reference is missing.
L 143: “stimulation time” ? meaning?
Author Response
Overall the presentation is too wordy, not to the point. The right scientific wordings have not been used to explain the technical aspects. The manuscript requires significant technical reformatting, systematic presentation approach. The English language, readability needs significant improvement.
Thank you for your appreciable recommendation. We used in the latest version of the manuscript the English service of MDPI.
Discussion and conclusion is a bit haphazard and jumbled. This needs a more systematic, organized scientific approach.
Thank you for your kindly observation. The entire paper was deeply reviewed and several points were modified to improve the organization scientific approach.
The reader is lost about what exactly the authors are trying to discover or invent.
Thank you for your comment. We think that is clear by line 8 to 11 which establish: The main result of this study is the selection of a combustion chamber height for the reference calorimeter that produces the best performance in the combustion process, which is 70 mm, as well as the main differences in using a three-step mechanism and a skeletal model to simulate an oxy-fuel combustion reaction.
Line 1: “effect of height” of what?
Maybe it could be confusing, to avoid it, we change the sentence by: This paper focuses on the numerical modeling of the effect of the height of a combustion chamber on the development of a reference calorimeter whose objective is to measure the calorific value of natural gas.
L 7: “woks” ?
Thank you for your kindly observation, that is a huge mistake by us. It has been changed.
Lines 63-70, too much text is spent on explaining the advantages of CFD.
Thank you for your kindly comment. We delete some lines and keep the necessary information to introduce the CFD use in our work.
Figure 2 caption, reference is missing.
Sorry for that mistake. However the PDF archive shows [6] which correspond to Haloua, F.; Hay, B.; Filtz, J. New French reference calorimeter for gas calorific value measurements. Journal of thermal analysis and calorimetry 2009, 97, 673–678. doi:10.1007/s10973-008-9701-z.
L 143: “stimulation time” ? meaning?
Sorry for that vocabulary, we want to mean the time required for the computer to solve our numerical model by the ANSYS software, but we have diced change it by ...computational time... in (line 141).
Round 2
Reviewer 1 Report
My concerns have been addressed by the author's revisions. The significance of the work is the use of CFD modeling to identify the physical parameters required to produced an accurate calorimeter. This is especially true for combustion reactions and designing to monitor the final product composition. The paper addresses these issues.
One minor point is Figure 8 which 3 curves and the legend only identifies two of them.
Reviewer 2 Report
The authors have made adequate revisions.